# Detection and Classification of Printed Circuit Boards Using YOLO Algorithm

Matko Glučina [ID], Nikola Anđelić *[ID], Ivan Lorencin [ID] and Zlatan Car [ID]

Department of Automation and Electronics, Faculty of Engineering, University of Rijeka, Vukovarska 58, 51000 Rijeka, Croatia
* Correspondence: nandelic@riteh.hr

**Abstract:** Printed circuit boards (PCBs) are an indispensable part of every electronic device used today. With its computing power, it performs tasks in much smaller dimensions, but the process of making and sorting PCBs can be a challenge in PCB factories. One of the main challenges in factories that use robotic manipulators for "pick and place" tasks are object orientation because the robotic manipulator can misread the orientation of the object and thereby grasp it incorrectly, and for this reason, object segmentation is the ideal solution for the given problem. In this research, the performance, memory size, and prediction of the YOLO version 5 (YOLOv5) semantic segmentation algorithm are tested for the needs of detection, classification, and segmentation of PCB microcontrollers. YOLOv5 was trained on 13 classes of PCB images from a publicly available dataset that was modified and consists of 1300 images. The training was performed using different structures of YOLOv5 neural networks, while nano, small, medium, and large neural networks were used to select the optimal network for the given challenge. Additionally, the total dataset was cross validated using 5-fold cross validation and evaluated using mean average precision, precision, recall, and F1-score classification metrics. The results showed that large, computationally demanding neural networks are not required for the given challenge, as demonstrated by the YOLOv5 small model with the obtained mAP, precision, recall, and F1-score in the amounts of 0.994, 0.996, 0.995, and 0.996, respectively. Based on the obtained evaluation metrics and prediction results, the obtained model can be implemented in factories for PCB sorting applications.

**Keywords:** classification; detection; PCB; semantic segmentation; YOLOv5

## 1. Introduction

In today's age of modern technology, all electronic devices contain at least one printed circuit board (PCB). PCBs are a platform that, using electronic components such as semiconductor chips and capacitors, enables the interconnection of electronic system components, guarantees signal quality, timing, and many other valuable functions, and for this reason, it can be found as a part of the following:

- Medical devices—such as computed tomography (CT) infusion pumps [1];
- Consumer electronics—many refrigerators, computers or communication devices [2,3];
- Automotive components—control systems or sensors [4–6];
- Maritime applications—in communication systems or navigation systems[7];
- Military and defense applications—instrumentation for monitoring threats, a control system for jamming radar systems [6,8] and so on.

However, the process of producing a single PCB is quite complex and requires a large investment in equipment, and the production process consists of more than 50 steps [9]. Apart from the complex way of making the PCB itself, there is an additional problem: the PCB industry exposes workers to high doses of toxic metals, solvents, acids, and photolithographic chemicals. For decades, intensive chemicals and careless manufacturing processes exposed thousands of workers to reproductive toxicants and carcinogenic chemicals. Due

to health-related issues of the production and automation of some processes, many factories underwent modernization [10]. Many funds were invested in the health of the workers themselves, but also in speeding up the PCB production process. With this in mind, the goal of this research is to produce an AI model that, based on input data (images) obtained from a camera, can acquire output information about the location, class, and segment of the object, in this case, PCB. At the same time, it must meet the high-performance requirements with the aim of implementation in the robotic systems that use segmentation characteristics to perform the task (in this matter, it is sorted depending on the PCB class). For this reason, a recently developed YOLOv5 instance segmentation algorithm will be tested, which, in addition to detection and classification so far, has a segmentation property that gives the robot the ability to segment the PCB class and trace it perfectly. It is important to emphasize that the term "instance segmentation" is used for algorithms that segment multiple classes on one image; however, in this research, this is not the case because one PCB class is segmented on one image, and it is better to use the term semantic segmentation instead of instance segmentation. The influence of the network size and the level of reliability of detection and segmentation of trained models will also be investigated for the problem of objects with similar characteristics at a higher production speed (on the production line in the PCB manufacturing process). Obtaining a high-quality model trained with the YOLOv5 algorithm creates an opportunity to solve the problem of uneven sorting, i.e., multiple classes independent of the order on the production line. The robotic manipulator will almost perfectly recognize any of the 13 classes of PCBs and, based on the recognized class, place them in the designated section, regardless of the PCB class order on the conveyor belt. This speeds up and improves the sorting process and the sorting problem with a very simple implementation into the robotic system.

The use of image-based machine learning (ML) methods for various tasks regarding PCB has been the subject of much research in the past several decades. Lu et al. (2022) [11] focused on the creation of an automated system for sorting electronic components that were detached from PCBs. The authors applied the YOLOv3 algorithm for the classification and localization of the PCBs. Satisfactory quality and speed for application in real-time scenarios were achieved with the presented approach. YOLOv3 has also been applied to the problem of defect detection by Wang et al. (2022) [12], who applied it using the darknet backbone. The authors managed to achieve a detection success rate of 89.3% on a relatively small sample size of 2000 images. Defect detection was also the topic of research by Ling et al. (2022) [13], who developed a custom defect detection framework based on a deep Siamese segmentation network. The authors achieved significant results with the highest classification rate of 93.49% and mIoU detection precision of 82.94%. Bhattacharya and Cloutier (2022) [14] also created a custom framework for fault detection, comparing it to various networks, including YOLO and faster region-based convolutional neural network, achieving mAP of 98.1%. Some of the authors focused on individual component classification, such as Ndayishimiye et al. (2021) [15] and Yoon and Lee (2021) [16]. The authors both applied YOLO networks—versions 3 and 2, respectively. Both papers show satisfying results with a classification performance of mAP of 99.23% for the former and an accuracy rate of 99.86% for the latter. From the research in the area, it is obvious that YOLO-based detection frameworks have common use in detection/localization tasks. Still, certain points have not been addressed in recent research. First is the instance segmentation aspect—most of the research has focused on object detection and localization. This approach does not provide information on the orientation of the device, which can cause issues in instances when the detected objects need to be precisely manipulated, e.g., using an industrial robotic manipulator for sorting [17,18]. Another aspect that is lacking in recent research is the focus on the entire PCB classification/localization, with most of the focus being given to the classification of individual SMD parts and/or faults. The detection of entire PCBs can serve multiple purposes, but the main one, especially when considering instance segmentation, is task allocation for automated sorting [19]. Yang et al. (2022) [20] demonstrated a CNN system for fruit sorting using CNNs, which achieved a classification

accuracy of 99.83%. Similar efforts were demonstrated by Wu et al. (2021) [21] who used a segmentation framework based on DeepLabV3+ for hydroponic object sorting and by Li et al. (2022) [22] for parcel sorting via CNN. Both papers demonstrate high accuracy with 99.24% and 97.00%, respectively. Interestingly, special focus has been given to waste management and recycling [23]. Kumar et al. (2021) [24] demonstrated an application of the SVM classifier, achieving 96.5% accuracy on the task of COVID waste management. Ku et al. (2021) [25] focused their research on automated grasping and sorting of construction and demolition waste using a robotic manipulator. The research demonstrates that the application of deep learning methods can improve the performance of such a system by 20%. While many authors noted the issue of electronic waste management [26–28], with some even discussing the possible application of AI [29,30], there is a lack of papers focusing on the direct application in recent years. Finally, there is a lack of research on the performance of the newer variants of the extremely popular YOLO algorithm, such as version 5 [31,32], which is generally shown to introduce better performance qualitatively, as well as speed-wise [33,34].

In addition to the experimental testing of the recently developed algorithm, the following hypotheses will be tested:

- Which type of YOLOv5 network model meets the given requirements for performing the presented task?
- Does the size of the used backbone model influence the results?
- Is there a possibility of implementing the obtained model for real purposes?

This paper is divided into three sections. The first is materials and methods, in which the used dataset will be explained in terms of the classes it consists of and the type of dataset used. In addition, the methodology description, explanation of the structure of the YOLOv5 algorithm, as well as the networks used to obtain the model and the associated hyperparameters used to obtain each model will be given. In the second section, the results obtained from this research will be presented and discussed, and the optimal model that can be applied for industrial purposes will be defined. In the last section, the reached conclusion will be presented. It will also be pointed out whether the proposed AI algorithm solved the given task, and the recommendations in future works for further development regarding this research thesis.

## 2. Materials and Methods

This section describes the dataset used in this research, the way of labeling images, and the preprocessing method that was used to improve the robustness of the model itself. Additionally, the method used for the detection and segmentation of objects, the models used, and the configured hyperparameters together with other settings that were included in the development of the final model are described.

### 2.1. Dataset Description and Preparation

The dataset used in this research consists of 13 classes of PCB microcontrollers and is publicly available on the Kaggle repository [35]. It consists of 1300 images (from the original 8125) that were created by applying different actions such as rotations, translations, centering, etc. Furthermore, the resolution is not consistent and uniform; it consists of different resolutions, and the mean resolution value is 1949 × 2126 pixels by height and width. It is noticeable that the total dataset is reduced by 6825 images, and this is not a coincidence. A CNN-based algorithm has been shown to easily achieve satisfactory performance on large datasets [36]. To investigate whether the model can be trained on a smaller set, and thereby speed up the data collection process in a possible application, a limited dataset is used. For this reason, a condensed dataset is formed that has at least one of the actions shown in Table 1.

**Table 1.** Description of the original dataset together with the associated generation actions.

| Microcontroller Class | Number of Images | Applied Dataset Generation Methods |
|---|---|---|
| Raspberry Pi A+ | | |
| Arduino Mega 2560 (Blue) | | |
| Arduino Mega 2560 (Black) | | |
| Arduino Mega 2560 (Black and Yellow) | | wide left rotation, |
| Arduino Due | | shallow left rotation, |
| Beaglebone Black | | neutral rotation, |
| Arduino Uno (Green) | 1300 images, 100 for each class | shallow right rotation, |
| Raspberry Pi 3 B+ | | wide right rotation, |
| Raspberry Pi 1 B+ | | 12 inches left of the camera position, |
| Arduino Uno Camera Shield | | 6 inches left of the camera position, |
| Arduino Uno (Black) | | centered horizontally relative to the camera, |
| Arduino Uno WiFi Shield | | 6 inches right of the camera position, |
| Arduino Leonardo | | 12 inches right of the camera position |

The background on which the dataset was generated is a white-to-gray gradient, which reduces the complexity of training the AI algorithm and practically represents ideal conditions; however, to achieve the system substandard (such as shadows, non-functional operation of one or more reflectors, or movement of the PCB itself in a non-ideal position), augmentations and processing methods were applied, which is described later in the paper. To show the actions and the general appearance of the dataset, several examples of PCBs are shown in Figure 1.

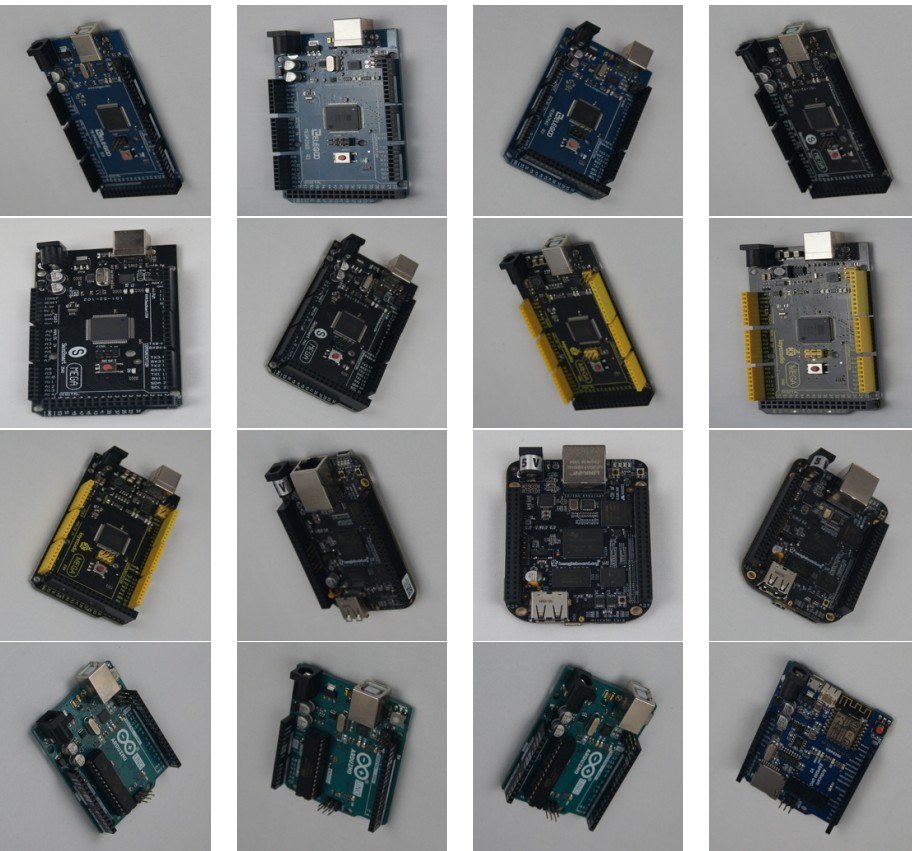

**Figure 1.** Display of the generated dataset and the environment during its creation.

Furthermore, it is important to note that the entire dataset was additionally modified, i.e., it was subjected to additional processing methods. So in this case, the first action is that absolutely all images of the new dataset are subjected to image resolution matching.

In the new dataset, all images are reduced to 640 × 640 resolution. The first reason is that the training process itself is performed much faster. Secondly, the dataset is quite trivial, meaning it consists of one microcontroller and the background, and there are no additional smaller objects that confuse the algorithm during training. The third reason is the use of pre-trained values that will be described later in the next section [37,38]. The last reason is to avoid distortion of the image in the case of increasing the resolution from a smaller volume to a larger one, for example, in the case of increasing the image resolution to 1920 × 1920, information may be lost when training the algorithm. The initial dataset is not annotated, so to perform localization, detection, and segmentation using the YOLOv5 algorithm, polygonal annotation is required. In this research, every one of 1300 images was annotated using Roboflow annotation software [39], and one of the annotation examples is shown in Figure 2.

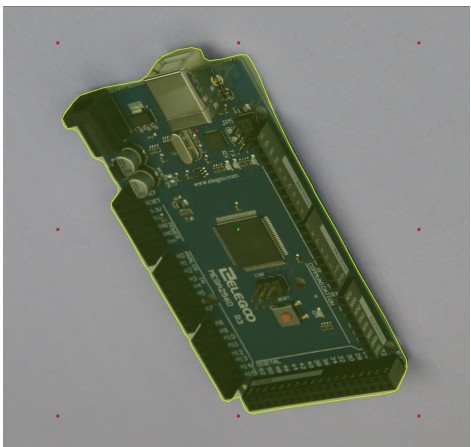

**Figure 2.** Labeling images for YOLOv5 semantic segmentation algorithm using Roboflow smart polygonal tool.

The main reason for choosing Roboflow annotation software is its simplicity and affordability in annotating images. By using the "Smart Polygon Labeling" option, a vastly larger and more precise number of annotation points surrounding the desired object are obtained, as seen in Figure 2.

Each point annotated using the Roboflow annotation software is written in TXT format, which is later used to train the YOLOv5 algorithm, but it is noticeable that the record is different than the previously defined width and height image. This is because the YOLO algorithm uses normalized pixel values. With this step, the preparation of the new dataset is done, the only thing left is to prepare the dataset for cross-validation (CV). As there is currently no automatic fold creation tool, the CV was manually performed using a five-fold CV.

The total dataset is divided into five equal parts (Fold 1–Fold 5) which makes 20% of the data used for each fold. Five separate datasets (Split 1–Split 5) are then created, consisting of one fold, i.e., the test set, and the other four folds that form the training set. For example, for the first split (Split 1), Fold 1 is the test set, while the other four (Fold 2–Fold 5) are grouped and used for the training set. For the second split (Split 2), Fold 2 is used for the test set while the other folds (Fold 1, Fold 3, Fold 4 and Fold 5) are grouped and form the training set. By applying this procedure to the other three splits, five separate datasets are obtained with the same amount of images (1040 for training and 260 for testing), but with different data (images) for each split separately. By using this methodology, the problem of overfitting is avoided, and a more precise evaluation of the AI algorithm used is carried out.

With this step, the description and preparation of the dataset are completed. In the next section, the YOLOv5 algorithm for semantic segmentation is additionally described, which is used to solve the problem of detection and segmentation of PCB micro-controllers [40].

*2.2. Methodology*

As presented in the Introduction of this article, there is not much literature on semantic models used to identify specific objects. The robotic manipulator can easily perform "pick and place" tasks, but the problem arises in cases of non-ideal objects on the conveyor belt, excessive movement, or densely stacked, randomly distributed, reflective, shiny, and plastic-wrapped objects [41–43]. For this reason, the development teams are trying to find a way in which it can be detected and localized in a fast and efficient way, and then determine the form in which the robotic manipulator can grasp the object with quality without too much difficulty. Semantic segmentation is a newer method, also known as image segmentation, and is part of computer vision methods in which the computer recognizes the object in the image along with its contours. It is an upgrade to conventional object detection in which each prediction, together with the bounding box, includes a shape with a common center, width, and length. The usage of this method enables determining the number of objects in the image, classification, and outline, which is the main advantage of using it. In cases where the size of an identified object must be measured, the application of the method has shown useful [44,45], and for that reason, it represents the most suitable solution for the proposed problem of sorting PCBs in factories.

Figure 3 shows the structure of the YOLOv5 network, which is composed of three parts: backbone, neck, and head (output result). The backbone is located right after the input image, and it serves in feature extraction. After that, the data are processed with the neck of the CNN, which performs resolution feature aggregation. Finally, the final predictions based on the resolution of the object are generated in the head [46,47]. In the backbone part, the input image with a resolution of $640 \times 640 \times 3$ passes through the focus layer (the layer used for the transformation from space to depth). After the initial transformation, with the help of the slice operator, the image is converted to a size of $320 \times 320 \times 12$ and becomes a map of features. By passing through the convolution operator of 32 convolution kernels, it finally becomes a $320 \times 320 \times 32$ feature map. The convolutional module CBL represents the sum of the convolutional kernel that convolutes with the input layer, which produces a tensor of outputs (Conv2D), batch normalization process (BatchNormal), and LeakyRELU activation function [48]. BottleneckCSP is used to extract features from a feature map. Compared to other large-scale CNNs, the BottleneckCSP structure can reduce the duplication of information gradients in the optimization part of the neural network process. That represents one of the key parts of the entire YOLOv5 network structure because its parameters occupy the largest part of the parameter amount of the entire network [49]. When changing the width and depth of the BottleneckCSP part of the network, the basic idea of this research also emerges, namely, four YOLOv5 models: small network (YOLOv5s), medium-sized network (YOLOv5m), large network (YOLOv5l) and very large network (YOLOv5x) [31,48]. The SPP module serves to increase the receptive field of the network and acquires additional network features of different scales. Additionally, YOLOv5 contains a bottom-up pyramid construction based on the feature pyramid network (FPN) structure [31,50], which is visible in Figure 3. The FPN layer plays a very important role because it transmits semantic features from top to bottom, and the feature pyramid itself contributes to the robustness of the extracted bottom-up positioning features. The feature is a combined aggregation from different layers of extracted features to further improve the performance of the network and thus the ability to detect targets at different scales. This is visible near the end of the image where the classification results and object coordinates are displayed.

As previously defined and explained, the influence of the size of the network has a large role in the outcome of the results, thus opening up the possibility of testing the given theory to solve the initially defined challenge. Therefore, in the continuation of this research, the influence of YOLO5s, YOLO5n, YOLO5m, and YOLO5l on the results of detection, classification, and segmentation of PCB micro-controllers is examined.

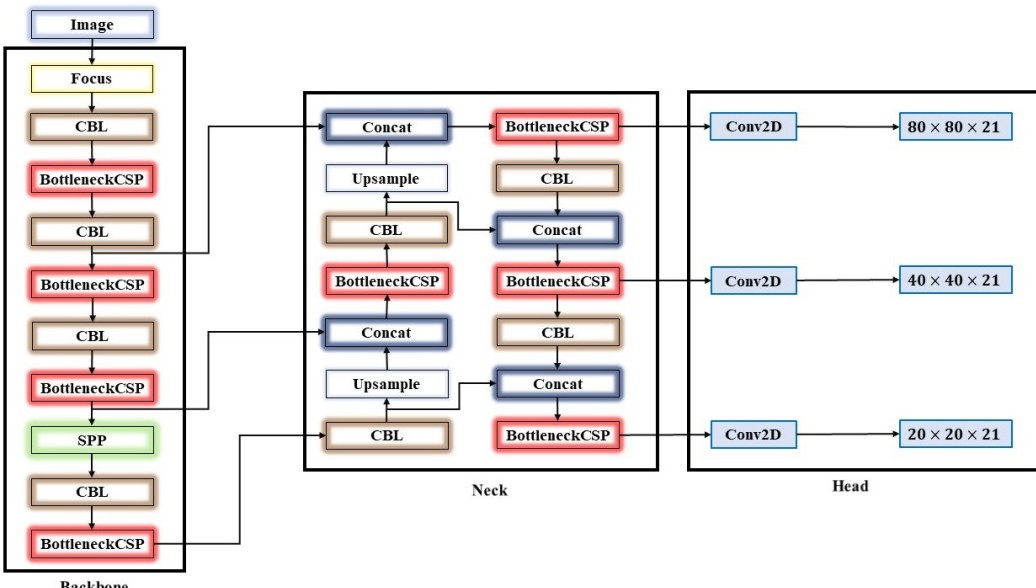

**Figure 3.** The YOLOv5 network framework.

Configuration of the YOLOv5 Semantic Segmentation Algorithm

The training specifications of the YOLOv5 semantic segmentation algorithm used in this research can be seen in Table 2.

**Table 2.** YOLOv5 training parameters used in this research.

| Weights | Network Model | Hyperparameters | Epochs | Batch Size | Image Size | Patience | Intersection over Union Threshold |
|---------|---------------|-----------------|--------|------------|------------|----------|-----------------------------------|
| yolov5n-seg | yolov5n-seg | lr0: 0.01 lrf: 0.01 momentum: 0.937 weight_decay: 0.0005 | 1000 | 30 | 640 × 640 | 100 | 0.6 |
| yolov5s-seg | yolov5s-seg | | | | | | |
| yolov5m-seg | yolov5m-seg | | | | | | |
| yolov5l-seg | yolov5l-seg | | | | | | |

In Section 2.1, the method of performing CV is described in terms of training the YOLOv5 algorithm. For each fold, the corresponding pair of weights and a network model with common hyperparameters, number of epochs, batch size, image size, and early stopping (patience) were developed. The intersection over union threshold (IoUt) is a parameter that shows the percentage of overlap between the predicted bounding box and the ground truth bounding box. In this research, its value was set to 0.6, i.e., 60%. This means that all predicted bounding boxes containing at least 60% of the initially labeled image are accepted as true positive predictions. In this way, 20 different models were developed with different subsets of data. The network model is the difference between the width and depth of the network structure described in the previous section, and their difference is visible in Table 3.

**Table 3.** Network structure specifications for different YOLOv5 models.

|  | YOLOv5n | YOLOv5s | YOLOv5m | YOLOv5l |
|--|---------|---------|---------|---------|
| depth_multiplier | 0.33 | 0.33 | 0.67 | 1.0 |
| width_multiplier | 0.25 | 0.5 | 0.75 | 1.0 |

From Table 3, the depth multiplier corresponds to what 'depth' the model will reach; in other words, it ultimately adds more layers to the neural network itself, while the width

corresponds to the number of filters in the layers, which results in more channels on the output layers. Weights are the selection of pre-trained values that are used for faster training compared to the "starch" approach. Each model is trained for 1000 epochs with early stopping if the results do not improve after 100 epochs at a resolution of 640 × 640. The batch size is the number of possible samples that are simultaneously processed before the model is updated, and it was set to 30 batches. In the hyperparameters column, four hyperparameters are defined as lr0, which indicates the initial learning rate, lrf, which represents the final OneCycleLR learning rate, and momentum is the accumulation of movement, i.e., how much of the previous value affects the further change of weight values. Weight decay is a regularization process that allows the ML algorithm to reduce model complexity and prevent potential overfitting.

Otherwise, the use of augmentation methods in practice is welcome. These were not originally mentioned for a unique reason because YOLOv5 has an integrated albumentation library [51], which is integrated into the algorithm itself. Accordingly, additional actions were applied to the dataset during training, shown in Table 4. Additionally, it is important to emphasize that the YOLOv5 algorithm implements the augmentation of images in the training process. This means that based on the data (images) found in the training set, augmentation is performed on them, while the test set remains untouched, provided that the data division was initially performed, which was done and described in Section 2.1. This approach avoids the problem of "data leaking", i.e., repetitions of identical data in the train and test sets.

**Table 4.** Metrics and processes applied to the dataset.

| | Method Applied | Value |
|---|---|---|
| 1 | hue augmentation (fraction) | 0.015 |
| 2 | saturation augmentation (fraction) | 0.7 |
| 3 | value augmentation (fraction) | 0.4 |
| 4 | rotation | ±0.15 |
| 5 | translation | ± 0.1 |
| 6 | scale | ±0.5 |
| 7 | flip left-right (probability) | 0.5 |
| 8 | mosaic (probability) | 1.0 |
| 9 | blur with a probability of applying the transform 0.5, with maximum Gaussian kernel size of 3 and 7 | |
| 10 | median blur parameters were set up identically as blur | |
| 11 | contrast limited adaptive histogram equalization (CLAHE) with the probability of 0.01 262 and upper threshold value for contrast limiting from 1 to 4.0 and tile grid size 8 × 8. | |

A short description of individual methods is as follows: hue augmentation is used to randomly change the colors of the input image channels, so the result is the ability of the model to consider alternative color schemes and scenes on input images. Saturation augmentation increases the saturation of an image and it has similar characteristics as hue augmentation, except that it adjusts how vivid the image is. Value augmentation is the fraction of augmented images while augmenting the dataset for hue and saturation augmentation. Rotation is the augmentation method that rotates the image to a certain degree. Similar to rotation, translation translates images in different directions. The scale parameter is used to scale the image by a certain percentage. Flip serves to reverse the image with the option defined in a certain percentage. Mosaic is a quality option that serves to train a neural network that works in a way similar to tiling, by four source images and combining them into one. Blur and median blur are similar techniques, which are used to cause the blurring effect to accomplish that the image is not ideal. CLAHE is a

variant of adaptive histogram equalization (AHE), which takes care of over-amplification of the contrast.

Accordingly, the entire work methodology can be reduced to one graphic representation, which is shown in Figure 4.

By dividing the data into five folds, we obtain five different subsets of data that were pre-processed using resize. Afterward, each fold is trained four times using one of the offered models together with the corresponding pre-trained weights. During training, the dataset is additionally subjected to the augmentation described in this section. Finally, at the end of the training, each model is exposed to evaluation methods and the results are attached, and the best, meaning the model which showed the best performance based on the evaluation metrics is selected. In the next section, the results of each fold are presented, and the individual metrics used to evaluate the performance of the model are described.

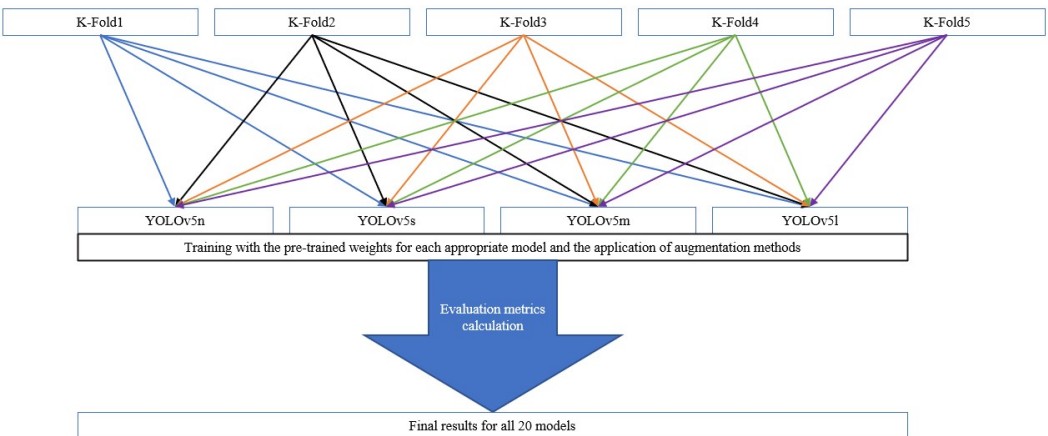

**Figure 4.** Abbreviated work methodology.

## 3. Results and Discussion

In this section, an overview of the results obtained from this research is given. Each model will be described with evaluation metrics that will also be described in this section. In the end, the results obtained from this research will be discussed, and the advantages and disadvantages of each network structure for the given challenge will be pointed out.

### 3.1. Verification of the Network Model

To evaluate the performance of the model, one of the key factors is the selection of evaluation metrics. So, in this subsection, the following metrics will be defined: mean average precision (mAP), precision, recall, and F1-score.

The mAP is used as a measure of computer vision model performance. The final result is equal to the mean value of the average precision metrics of all individual model classes. Its value ranges from 0 to 1, i.e., from 0% to 100%. The mathematical formula that describes the mAP [52,53] calculation can be shown as

$$mAP = \frac{1}{n} \sum_{k=1}^{k=n} AP_k. \tag{1}$$

In Equation (1), $AP_k$ stands for the average precision of the individual class, while n is the total number of the classes. The sum of all mean precision values for each class divided by the total number of classes results in the final mAP value. The precision metric, as the name suggests, determines the precision with which the model determines the class of the object and is calculated using a mathematical expression [54]:

$$precision = \frac{TP}{TP + FP}. \tag{2}$$

In Equation (2), TP refers to true positive prediction, while FP is false positive prediction. TP and FP are the numbers of correct positive and false positive predictions. More precisely, the number of TP divided by the sum of TP and FP represents the precision of the model expressed in the range from 0 to 1. Recall shows how many times the model found a positive case while training the algorithm, and can be calculated using

$$\text{recall} \ = \frac{\text{TP}}{\text{TP} + \text{FN}}. \tag{3}$$

Using Equation (3), it is possible to calculate the recall metric for a particular model. The only difference from Equation (2) is that false negative (FN) is used in the denominator instead of FP, and the final result is also on a scale from 0 to 1. The last in the series is the F1-score metric, which is the ratio of the harmonic mean of precision and recall from a given classifier. The mathematical description of F1-score is given by Equation (4), where the final result is the ratio of the product recall and precision divided by their sum and finally twice multiplied:

$$\text{F1-score} \ = 2 \times \frac{\text{recall} \ \times \ \text{precision}}{\text{recall} \ + \ \text{precision}}, \tag{4}$$

To evaluate the performance of the model network, the mean value ($\bar{X}$) of each metric is shown using the following equation [55]:

$$\bar{X} = \frac{1}{n} \sum_{i=1}^{i=n} X, \tag{5}$$

Equation (5) is identical to the mathematical formula defined for mAP. The only difference is that $\bar{X}$ is used as a universal variable for all individual metrics, i.e., precision, recall, mAP, and F1-score. The variable $X$ indicates a value of a specific variable, while $n$ is the number of used samples.

The last in the series of selected metrics is the standard deviation ($\sigma$), which represents the average amount of variability in the dataset used. The result $\sigma$ tells how much each score lies from the mean value. The given metric is shown as

$$\sigma = \sqrt{\frac{\sum (X - \mu)^2}{N}}, \tag{6}$$

where:

- $X$ each considered value;
- $\mu$ mean value of the considered population;
- $N$ is the number of values in the considered population.

The definition of all the individual metrics is an introduction to the evaluation of the individual obtained model. The only thing left is to present the obtained results, which is done in the following subsections.

### 3.1.1. Results for YOLOv5n Model

Table 5 shows the training results of the YOLOv5n semantic segmentation algorithm for localization detection and segmentation of PCBs. For each fold, i.e., each subset of data, performance metrics were calculated for the semantic part, i.e., the segmented mask (M) of the tile itself and the usual bounding box (B). It can be seen that the second fold converged the fastest in as many as 360 epochs, with mAP in the amount of 0.992 and precision and recall for B in the amounts of 0.993 and 0.995. As for the metrics related to M, the results were identical. The best result was given by the model created by the first fold with mAP in the amount of 0.994 for B and 0.995 for M, while precision and recall for both cases were 0.997 and 0.996. The final result of the YOLOv5n model was 0.987 in favor of mAP (B) and 0.992 for (M), while their $\sigma$ was 0.0019 and 0.0025 for the corresponding case.

**Table 5.** Results for YOLOv5n model.

| Fold | Early Stopping at | Precision (B) | Recall (B) | mAP (B) | Precision (M) | Recall (M) | mAP (M) | F1-Score (B) | F1-Score (M) |
|---|---|---|---|---|---|---|---|---|---|
| 1 | 949 | 0.997 | 0.996 | 0.994 | 0.997 | 0.996 | 0.995 | 0.997 | 0.997 |
| 2 | 360 | 0.993 | 0.995 | 0.992 | 0.993 | 0.995 | 0.992 | 0.994 | 0.994 |
| 3 | 630 | 0.997 | 0.994 | 0.991 | 0.997 | 0.994 | 0.991 | 0.995 | 0.995 |
| 4 | 368 | 0.996 | 1 | 0.993 | 0.996 | 1 | 0.993 | 0.998 | 0.998 |
| 5 | 503 | 0.996 | 0.996 | 0.990 | 0.996 | 0.996 | 0.990 | 0.996 | 0.996 |
| $\bar{X}$ | | 0.995 | 0.996 | 0.987 | 0.995 | 0.996 | 0.992 | 0.995 | 0.995 |
| $\sigma$ | | ±0.002 | ±0.002 | ±0.010 | ±0.002 | ±0.002 | ±0.001 | ±0.001 | ±0.001 |

### 3.1.2. Results for YOLOv5s Model

The results for YOLOv5s have similar values as for the previous case and are presented in Table 6. The fastest converging model was related to the third subset of data, i.e., the third fold. The change in metric values after 273 epochs was not significant. The model achieved results of precision, recall, and mAP related to B in the amounts of 0.996, 0.996, and 0.99, while for B it achieved practically identical results. F1-score was 0.996 in both cases. The best result was given by the first fold with mAP in the amount of 0.995, precision 0.996, and recall 0.996 for M, respectively, and 0.994, 0.996 and 0.996 for B. $\bar{X}$ of mAP for B and M were 0.992 and 0.991 with $\sigma$ of 0.001 and 0.002.

**Table 6.** Results for YOLOv5s model.

| Fold | Early Stopping at | Precision (B) | Recall (B) | mAP (B) | Precision (M) | Recall (M) | mAP (M) | F1-Score (B) | F1-Score (M) |
|---|---|---|---|---|---|---|---|---|---|
| 1 | 964 | 0.996 | 0.996 | 0.994 | 0.996 | 0.996 | 0.995 | 0.996 | 0.996 |
| 2 | 297 | 0.993 | 0.995 | 0.990 | 0.993 | 0.995 | 0.989 | 0.994 | 0.994 |
| 3 | 273 | 0.996 | 0.996 | 0.990 | 0.996 | 0.996 | 0.990 | 0.996 | 0.996 |
| 4 | 540 | 0.996 | 1 | 0.994 | 0.996 | 1 | 0.994 | 0.998 | 0.998 |
| 5 | 424 | 0.996 | 0.996 | 0.991 | 0.996 | 0.996 | 0.990 | 0.996 | 0.996 |
| $\bar{X}$ | | 0.997 | 0.996 | 0.992 | 0.996 | 0.996 | 0.991 | 0.996 | 0.996 |
| $\sigma$ | | ±0.001 | ±0.001 | ±0.001 | ±0.001 | ±0.001 | ±0.002 | ±0.001 | ±0.001 |

### 3.1.3. Results for YOLOv5m Model

The medium model YOLOv5m had similar results as the previous two cases, but there is a non-convergence of the results. In Table 7 in the first fold, unlike the previous two cases, there is no early termination of training, i.e., did not obtain a qualitative solution for the given problem. At the same time, in 1000 epochs, the model achieved precision, recall, and mAP in the amounts of 0.997, 0.998, and 0.995 for B, respectively, 0.997, 0.998, and 0.995, i.e., identical results, for M. F1-score was 0.997 for both cases. It is important to emphasize the training process was not completed because it needed more epochs to converge, but at 1000 epochs, it achieved the best results compared to the other four folds, as shown in Table 7. Furthermore, the $\bar{X}$ of precision, recall, mAP, and F1-score for B were 0.991, 0.993, 0.991, and 0.992 for M, respectively, and 0.991, 0.993, 0.992, and 0.992 for B. The variation of $\sigma$ was not significant and was in a range from 0.00213 to 0.01041 for the corresponding metric.

**Table 7.** Results for YOLOv5 medium model.

| Fold | Early Stopping at | Precision (B) | Recall (B) | mAP (B) | Precision (M) | Recall (M) | mAP (M) | F1-Score (B) | F1-Score (M) |
|---|---|---|---|---|---|---|---|---|---|
| 1 | 1000 | 0.997 | 0.998 | 0.995 | 0.997 | 0.998 | 0.995 | 0.997 | 0.997 |
| 2 | 331 | 0.973 | 0.988 | 0.989 | 0.973 | 0.988 | 0.991 | 0.980 | 0.980 |
| 3 | 490 | 0.993 | 0.984 | 0.990 | 0.993 | 0.984 | 0.991 | 0.989 | 0.989 |
| 4 | 583 | 0.996 | 1 | 0.994 | 0.996 | 1 | 0.993 | 0.998 | 0.998 |
| 5 | 506 | 0.996 | 0.996 | 0.990 | 0.996 | 0.996 | 0.989 | 0.996 | 0.996 |
| | $\bar{X}$ | 0.991 | 0.993 | 0.991 | 0.991 | 0.993 | 0.992 | 0.992 | 0.992 |
| | $\sigma$ | ±0.010 | ±0.006 | ±0.002 | ±0.010 | ±0.006 | ±0.002 | ±0.007 | ±0.007 |

### 3.1.4. Results for YOLOv5l Model

The last YOLOv5 semantic segmentation network model used in this research is YOLOv5l and its results are attached in Table 8. As in the previous case with YOLOv5m, the first fold did not converge, i.e., it was necessary to provide more epochs to achieve the possibility of converging results. The first fold was trained for 1000 epochs, and achieved results for precision, recall, mAP, and F1-score in the amounts of 0.997, 0.997, 0.995, and 0.997 for B, while for M, it achieved 0.997, 0.997, 0.995 and 0.997. The model trained on the fourth fold in 339 epochs converged the fastest with precision, recall, and mAP amounts of 0.994, 0.993, and 0.993 for B, while M achieved results in the amount of 0.99459, 0.993, and 0.993.

**Table 8.** Results for YOLOv5l model.

| Fold | Early Stopping at | Precision (B) | Recall (B) | mAP (B) | Precision (M) | Recall (M) | mAP (M) | F1-Score (B) | F1-Score (M) |
|---|---|---|---|---|---|---|---|---|---|
| 1 | 1000 | 0.997 | 0.997 | 0.995 | 0.997 | 0.997 | 0.995 | 0.997 | 0.997 |
| 2 | 440 | 0.992 | 0.995 | 0.990 | 0.992 | 0.995 | 0.988 | 0.994 | 0.994 |
| 3 | 411 | 0.928 | 0.931 | 0.987 | 0.928 | 0.931 | 0.988 | 0.929 | 0.929 |
| 4 | 339 | 0.994 | 0.993 | 0.993 | 0.994 | 0.993 | 0.993 | 0.993 | 0.993 |
| 5 | 547 | 0.996 | 0.996 | 0.989 | 0.996 | 0.996 | 0.990 | 0.996 | 0.996 |
| | $\bar{X}$ | 0.981 | 0.982 | 0.991 | 0.981 | 0.982 | 0.991 | 0.981 | 0.981 |
| | $\sigma$ | ±0.030 | ±0.029 | ±0.002 | ±0.030 | ±0.028 | ±0.002 | ±0.029 | ±0.029 |

### 3.2. Discussion

To clarify the final results of the research, the final results of the particular neural network model used will be additionally graphically presented.

A summary of all obtained mean values of individual metrics can be seen from the bar plot shown in Figure 5. The metric values of each model are defined with the YOLOv5 associated network. The term mean stands for $\bar{X}$ and std for $\sigma$, which were previously explained. Each model achieved results in the amount of 0.9 for $\bar{X}$, precision, recall, mAP, and F1-score, respectively. Although YOLOv5n had high performance, looking at the global level (concerning mAP(B) metric), it was the worst. Additionally, the $\sigma$ level has acceptably low bounds as shown in Figure 6.

The most metric deviations were in the case of the YOLOv5l model, which is visible in the bar plot of the large std. However, mostly the large model is used for detection and segmentation with smaller objects, such as small microchips and USB slot connectors. In this case, smaller models, such as nano and small, showed amazing performance, which shows that even smaller models that are quite faster and require less working memory

can perform the same, if not better, task, which greatly changes the nature of selecting models for training detection and segmentation of this type. All of the stated claims can be shown with examples of labeled and predicted classes that are shown in Appendix A, where Figures A1 and A3 represent real classes of PCBs, while Figures A2 and A4 are predicted classes.

**Figure 5.** Mean values of individual metrics for each YOLOv5 model.

In addition to the obtained metrics, the issue of integrating the obtained model into real situations is raised, and thus the size of the model itself and the necessary computational properties needed for detection and segmentation can be additionally described. Based on the mAP metric for M, the four best models that achieved the highest results were selected, and their memory size and the number of layers with the corresponding number of parameters were compared.

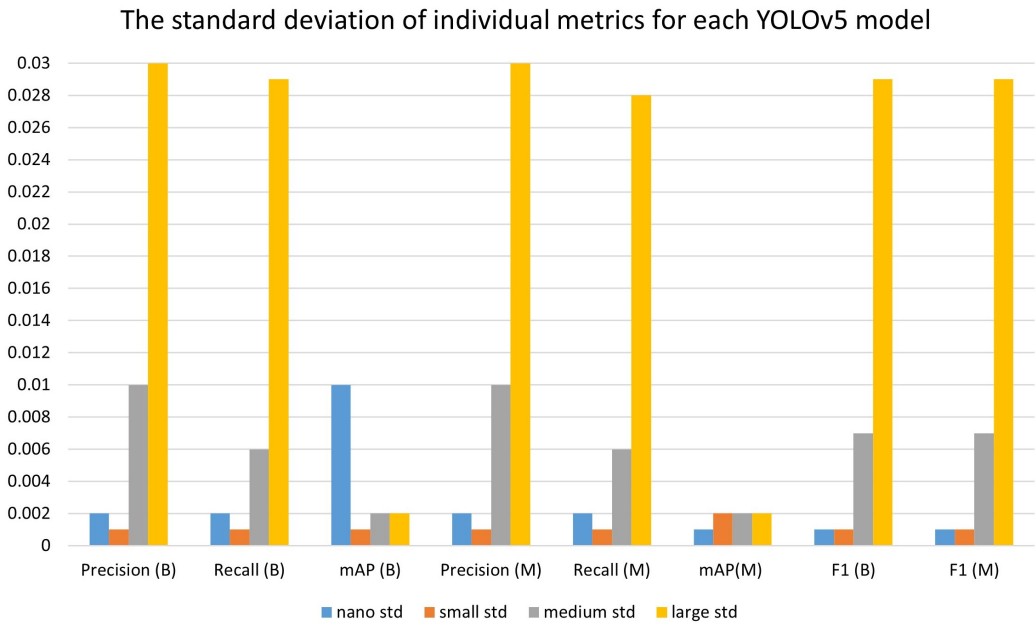

**Figure 6.** The standard deviation of individual metrics for each YOLOv5 model.

Regarding the selection of the best model shown in Table 9 for YOlOv5n, the first fold was selected for YOLOv5n, while for YOLOv5s, YOLOv5m, and YOLOv5l, the fourth models were selected. The YOLOv5n and YOLOv5s networks both have 165 layers, the more significant difference is in the number of parameters used to compute the results. The exported YOLOv5n model using the open neural network exchange (ONNX) format is slightly more than 4 megabytes (MB), the YOLOv5s model is about 23 MB, while the YOLOv5m and YOLOv5l are quite a bit larger, i.e., 85 and 186 MBs, which was expected based on their structure. Besides the memory and computational requirements, one of the most important features of the YOLOv5 algorithm is the inference speed in which YOLOv5 excels for each NN model. The inference time is calculated on the validation set (data that were not in the test set) and the mean value was taken for all five splits under the conditions that the IoU threshold was set to 0.6, while the confidence threshold was set to 0.9. As shown in Table 9 YOLOv5 has the lowest inference time using the YOLOv5s model. The YOLOv5n model is slower by 1 ms than the YOLOv5s model, while YOLOv5m is slower by 5 ms and YOLOv5l by as much as 9 ms. Regardless of the time difference, each model is within acceptable limits in terms of inference speed, which further confirms the possibility of applying the YOLOv5 algorithm AI for a given challenge. Based on the results given, which excel in every respect, the question of potential integration of the model arises, with the winner being the YOLOv5n and YOLOv5s models, which are significantly ahead of the other models with their memory size, metrics and inference speed . YOLOv5s model with relatively high mAP performance and small $\sigma$ is robust and almost perfectly recognizes all 13 classes of PCBs, and this trained model can be implemented in robotic systems for "pick and place" purposes, for example, on the production line in PCB sorting factories.

**Table 9.** Final concluding discussion and selection of a model for implementation.

| YOLOv5 Model | Memory Size in Kilobytes [KB] | Number of Layers | Number of Parameters | Mean Inference Speed in Milliseconds [ms] |
|---|---|---|---|---|
| YOLOv5n | 4079 | 165 | 1,895,986 | 12.575 |
| YOLOv5 | 29,481 | 165 | 7,430,786 | 11.175 |
| YOLOv5m | 85,242 | 220 | 21,700,850 | 16.225 |
| YOLOv5l | 186,171 | 275 | 47,533,602 | 20.651 |

## 4. Conclusions

This research presented the model size influence of YOLOv5 NN for the problem of semantic segmentation. Different hyperparameters were tested, and the results show that different hyperparameters influence different-sized model performance results. YOLOv5l is the most commonly used NN, but smaller networks demonstrate satisfactory performance as well, with up to six times smaller memory demand in the case of YOLOv5s, allowing for wider application on edge devices. The achieved results make it possible to address the initially posed hypotheses:

- YOLOv5 is capable of producing high-performance models for detection, localization, and segmentation, no matter the model size (YOLOv5n, YOLOv5s, YOLOv5m or YOLOv5l), with YOLOv5s providing results with the lowest $\sigma$;
- For the conditions present in the conducted research, the backbone of the model had no influence;
- YOLOv5 shows satisfactory classification performance, with F1 scores above 0.99, indicating possible use for sorting in production environments.

The conducted research was carried out on complete and undamaged PCB microcontrollers, and YOLOv5 satisfied the given requirements and enabled possible implementation in real situations. Several limitations of the paper should be addressed in future work. Due to the ideal conditions in which the dataset images were collected, it is important to note

that the application in a real environment may result in poorer performance. It is necessary to expand the dataset with images that are turned upside down, which are much more complex compared to the dataset used in this research. Additionally, it is recommended to test and, if necessary, retrain the YOLOv5 model on examples of multiple PCBs located in a single frame. To fully verify the performance of the model it should be integrated into an environment with a robotic manipulator to confirm and demonstrate that the developed model can perform well in a realistic environment.

**Author Contributions:** Conceptualization, N.A., I.L. and Z.C.; methodology, N.A., M.G. and Z.C.; software, Z.C., I.L. and Z.C.; validation, N.A., M.G. and Z.C.; formal analysis, N.A., M.G. and I.L.; investigation, N.A. and I.L.; resources, Z.C.; data curation, I.L., M.G.; writing—original draft preparation, M.G., N.A., I.L. and Z.C.; writing—review and editing: N.A., M.G. and I.L.; visualization, I.L. and Z.C.; supervision, I.L., N.A. and Z.C.; project administration, Z.C.; funding acquisition, Z.C. All authors have read and agreed to the published version of the manuscript.

**Funding:** This research received no external funding.

**Data Availability Statement:** The data used in this paper were obtained from a publicly available repository located at https://www.kaggle.com/datasets/frettapper/micropcb-images (accessed on 1 December 2022).

**Acknowledgments:** This research has been (partly) supported by the CEEPUS network CIII-HR-0108, European Regional Development Fund under the grant KK.01.1.1.01.0009 (DATACROSS), project CEKOM under the grant KK.01.2.2.03.0004, Erasmus+ project WICT under the grant 2021-1-HR01-KA220-HED-000031177, and University of Rijeka scientific grants uniri-mladi-technic-22-61, uniri-mladi-technic-22-57, uniri-tehnic-18-275-1447.

**Conflicts of Interest:** The authors declare no conflict of interest.

## Abbreviations

The following abbreviations are used in this manuscript:

| | |
|---|---|
| AHE | Adaptive Histogram Equalization |
| CLAHE | Contrast Limited Adaptive Histogram Equalization |
| CT | Computed Tomography |
| CV | Cross Validation |
| FN | False Negative |
| FP | False Positive |
| FPN | Feature Pyramid Network |
| IoU | Intersection Over Union |
| IoUt | Intersection Over Union Threshold |
| mAP | Mean Average Precision |
| ML | Machine Learning |
| ONNX | Open Neural Network Exchange |
| PCB | Printed Circuit Board |
| TP | True Positive |
| TN | True Negative |
| YOLOv5 | You-Only-Look-Once version 5 |
| YOLOv5n | You-Only-Look-Once version 5—nano |
| YOLOv5s | You-Only-Look-Once version 5—small |
| YOLOv5m | You-Only-Look-Once version 5—medium |
| YOLOv5l | You-Only-Look-Once version 5—large |

## Appendix A

In this part of the Appendix section, a presentation of the real class of PCBs (ground truth) is given, and a predicted class using the YOLOv5 model is shown.

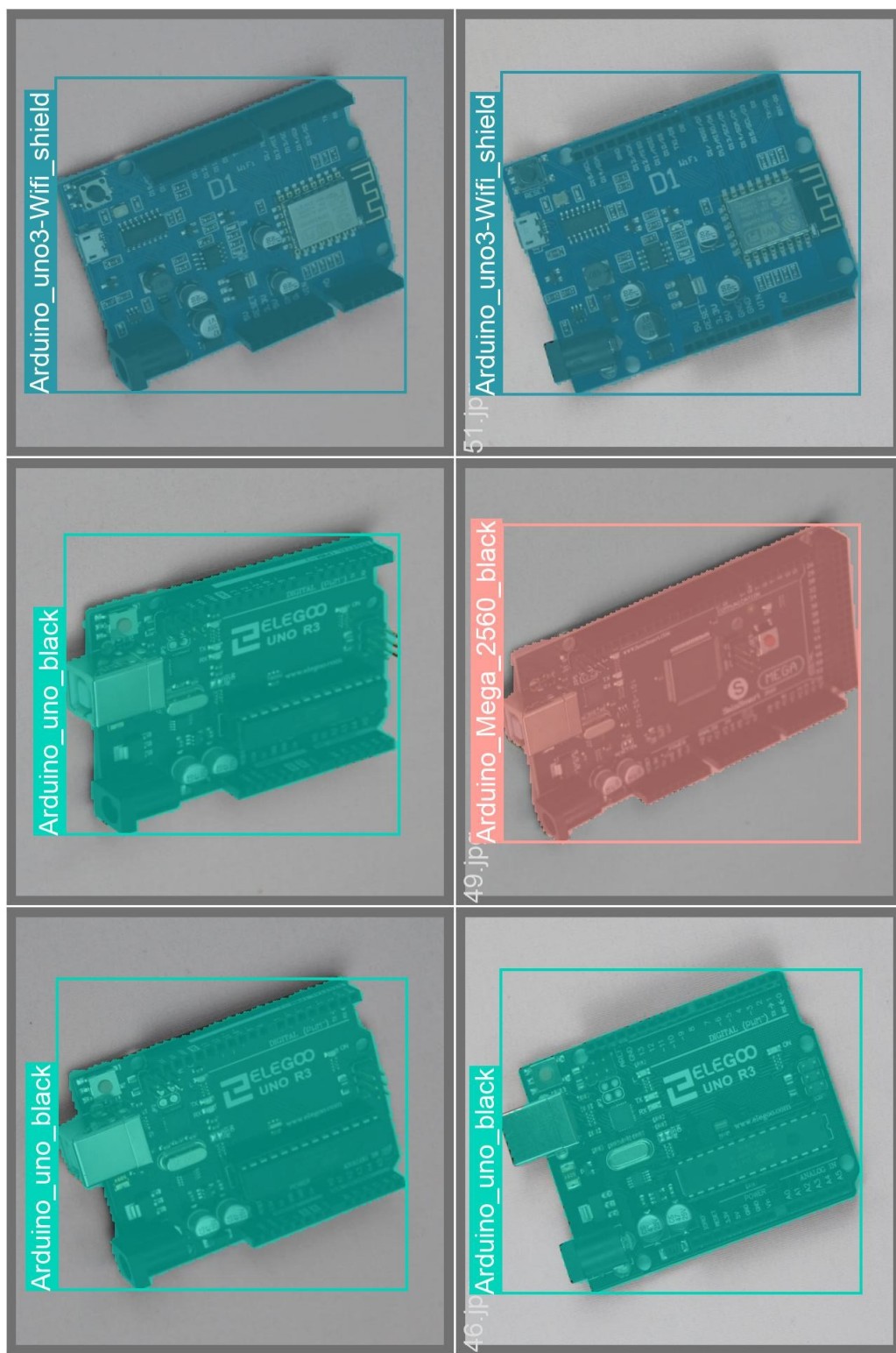

**Figure A1.** First validation of data marked with labels.

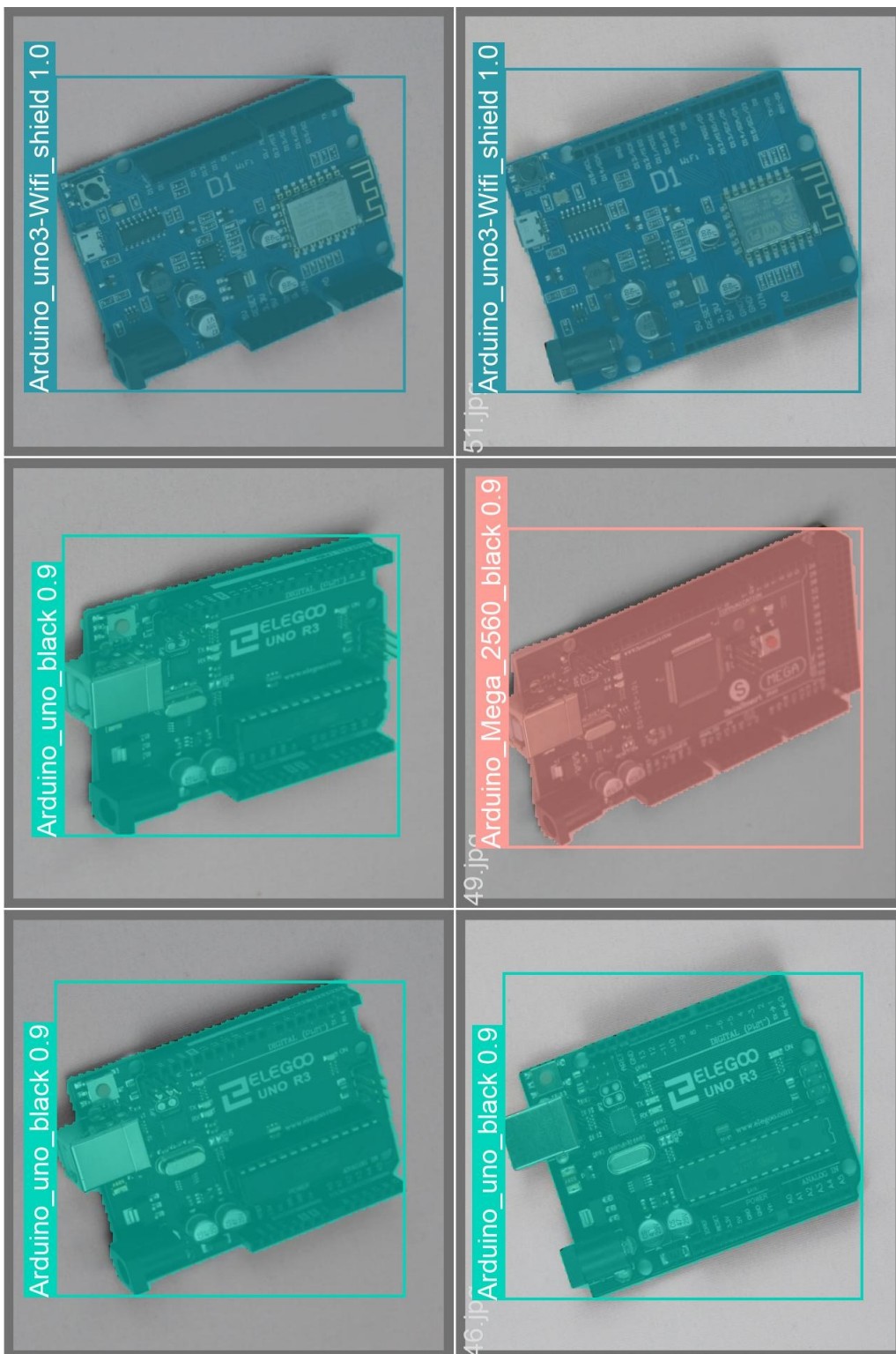

**Figure A2.** Intended classes of the first validation.

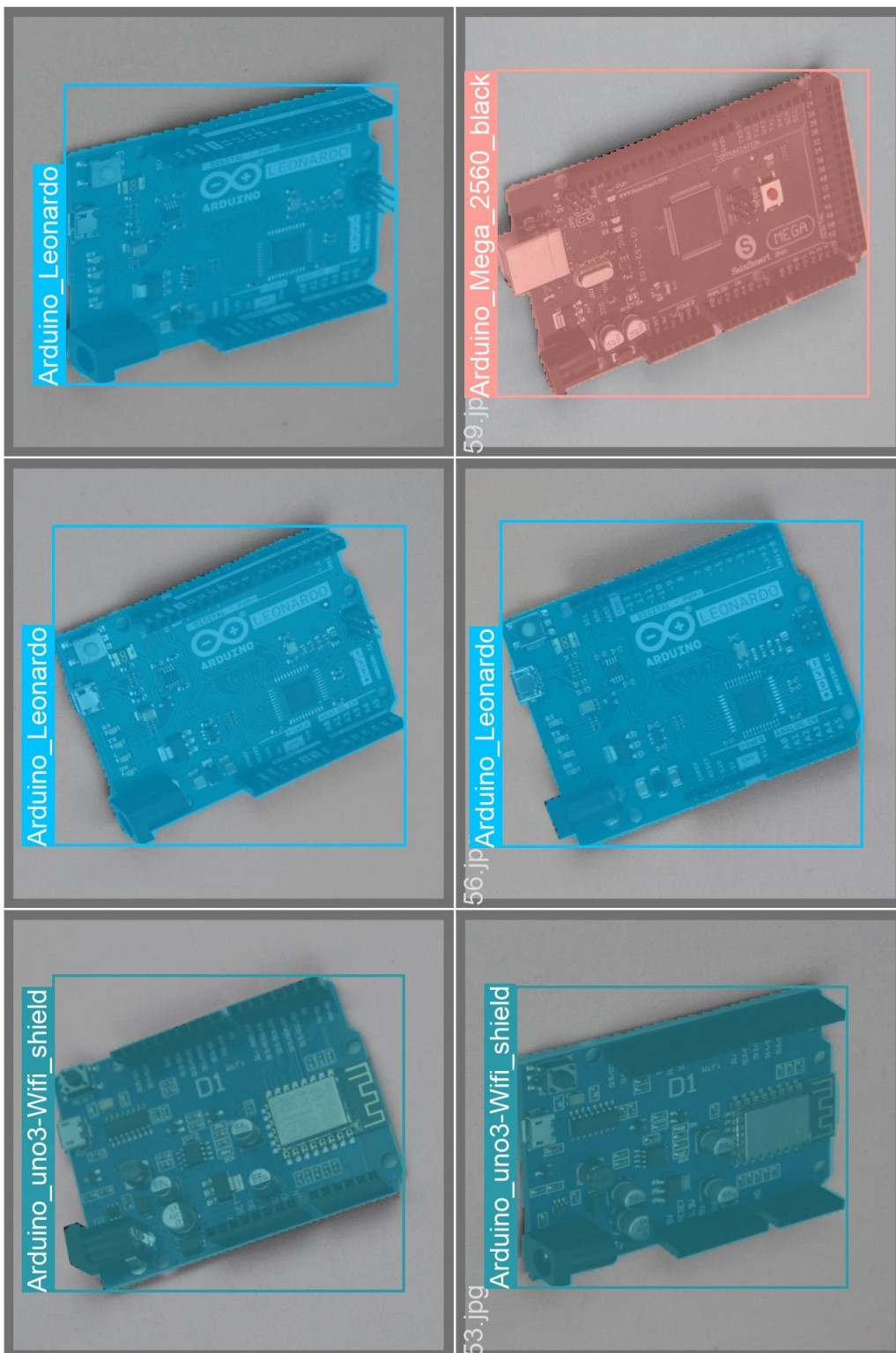

**Figure A3.** Second validation of data marked with labels.

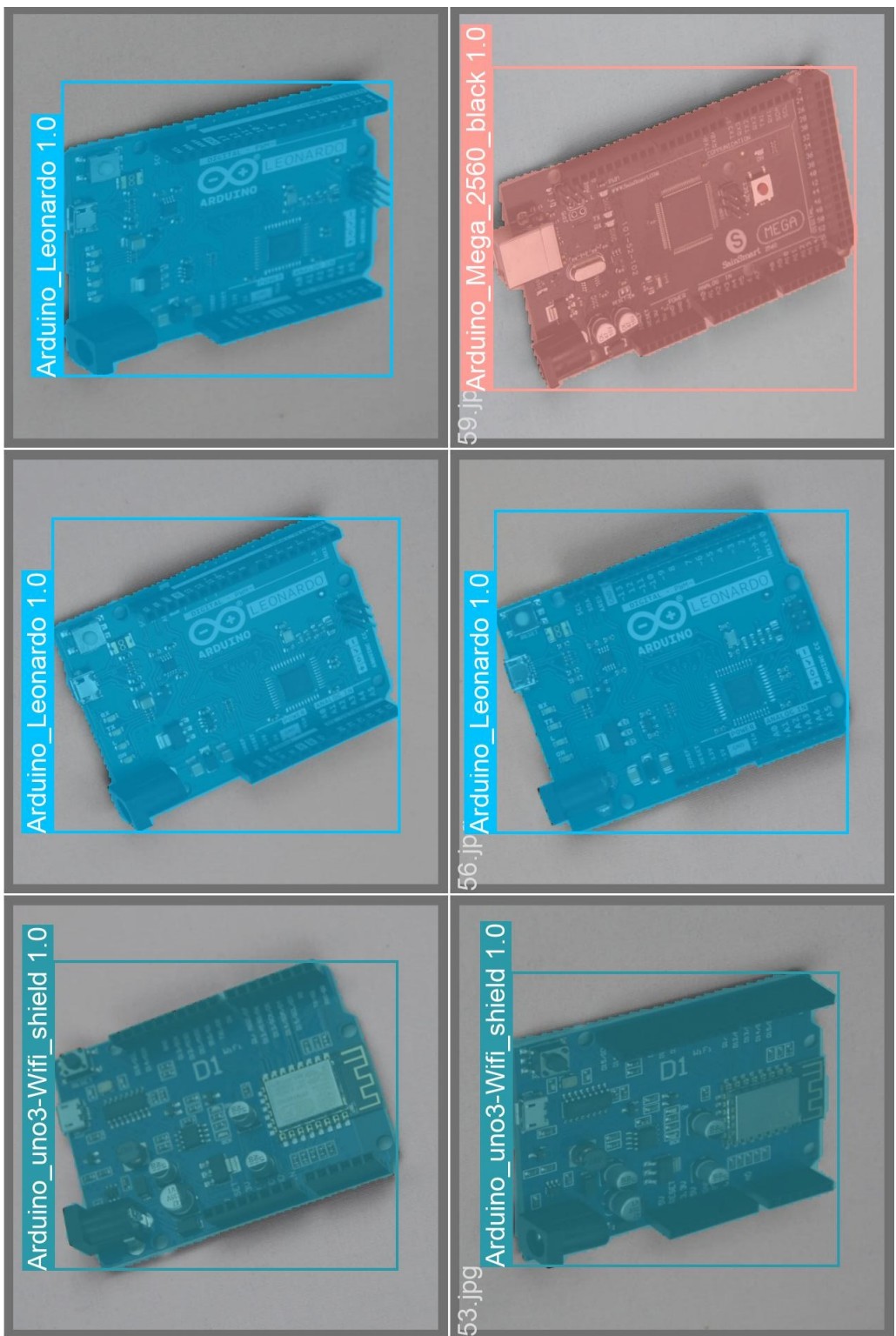

**Figure A4.** Intended classes of the second validation.

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
