# Peer review of "Detection and Classification of Printed Circuit Boards Using YOLO Algorithm"

_electronics, doi:10.3390/electronics12030667_

Round 1
Reviewer 1 Report
Review comments:
This manuscript entitled “Detection classification of printed circuit boards using YOLO algorithm” aimed to produce an AI model that will satisfy the accuracy requirements with the goal of implementation in robotic systems that uses segmentation characteristics to perform the task.
In the manuscript, the authors substantially show the presentation of the contributions and the experimental results of their proposed issues. In my opinion, the revised version of the manuscript is now acceptable for publication.
Author Response
Respected Reviever,
We thank you for your time and effort in reviewing our manuscript. The answers to the comments are posed below.
This manuscript entitled “Detection classification of printed circuit boards using YOLO algorithm” aimed to produce an AI model that will satisfy the accuracy requirements with the goal of implementation in robotic systems that uses segmentation characteristics to perform the task.
In the manuscript, the authors substantially show the presentation of the contributions and the experimental results of their proposed issues. In my opinion, the revised version of the manuscript is now acceptable for publication.
The authors thank you for your comments.
Kindest regards,
The authors
Reviewer 2 Report
The manuscript described an application of YOLOv5 for PCB boards identification. The manuscript is well-written and complete. The research is justified. Good literature review is provided in introduction.
The dataset used in the study, however, has extremely low difficulty for the PCB boards to be identified using an object detector. As a matter of fact, a simple color thresholding can easily segment out all the PCB boards in the images. In that sense, the application of YOLOv5 is meaningless and an overkill. This is why almost perfect results were achieved in the study. In 129-130, the authors mentioned “the background on which this dataset was created is completely white and there are no obstacles that can affect the training of AI algorithms”, which completely misses the point of AI of dealing with real life challenging tasks. The dataset is not suitable for evaluating the true capacity of YOLOv5.
I also wanted to comment on the “cross-validation” used in the study. The authors did not adopt a typical training-validation-test data split for model training and evaluation, which is simpler and less biased. Typically cross validation data will be exposed to the model during training, although which is not the case in this study and I need to comment on that too, therefore cross validation results are biased towards superior model performance.
In this study, however, the authors split the entire dataset into five smaller datasets, and trained five different models with the five smaller datasets that were further split into training and test datasets. I’m not sure whether this can be called “cross validation”, as no validation dataset is present and no validation was executed during model training. I believe the authors mistook their test datasets as validation datasets.
My final recommendation for this manuscript is rejection due to its very low novelty and research value. However, it does not hurt anything if the manuscript is published, purely for the objective information that it presents. Below are some of my other observations.
What is the IoU threshold for identifying TP/TN/FP/FN?
Line 133-155 provided unprofessional reasons for not using the entire original dataset. If using the entire dataset is unnecessary and would achieve similar prediction results, which is probably the case, then not using the entire dataset is justified. “Too much work” cannot be a reason for not going above and beyond in research.
Figure 1 should show the images with various dataset generation methods provided in Table 1. The current images in Figure 1 are rather similar to each other and do not seem to be representative.
Additionally, Table 1 is unnecessary if the entire original dataset is not used. Instead, only provide information of the shrunk dataset.
Table 2 and Figure 3 and their relevant content are redundant.
“Instance segmentation” is the wrong term to be used in this manuscript as there is not multiple instances in each image. “Semantic segmentation” should be used.
Line 378-380, be very careful with this statement as it is likely incorrect. “Deep networks are less accurate than shallow networks” does not make sense and is likely due to insufficient model training, i.e., too small of a dataset, too simple of a dataset. Deep network performance can plateau to the same level of shallow networks, but it is not because of the network itself, rather, it is because of the dataset.
Author Response
Respected Reviewer,
We thank you for your time and effort in reviewing our manuscript. The answers to the comments are listed below. In the manuscript, all changes are marked in blue.
“In 129-130, the authors mentioned “the background on which this dataset was created is completely white and there are no obstacles that can affect the training of AI algorithms”, which completely misses the point of AI of dealing with real life challenging tasks.”
Answer:
The given statement referred to the display of the environment in which the PCB is located, considering that the initial application and proposal for the use of the given algorithm was intended for production lines so that the robotic manipulator can "pick and place" PCBs. In most cases, the production conditions are ideal, ideal lighting, no influence of shadow or smoke, etc. As said, this is not the point of the AI algorithm itself, but to further improve the quality of detection, the augmentation method is described in the work. CLAHE affects contrast and noise and regulates low-contrast images, i.e. processes them to be more applicable to the purpose (with this, images with poor contrast are obtained with some disturbances such as shadows and brighter images, and both types are used for training the algorithm).
Blur and median blur are similar techniques, which are used to cause the blurring effect to accomplish that the image is not ideal.(As the name of the method suggests, blur is used for better training, this action additionally improves the training quality of the PCB dataset because it is not always ideal in which it is trained, like for example the original image.)
Furthermore, the given augmentations do not refer only to the marked object, but to the entire image, i.e. the training sample, which certainly contributes to the robustness of the detection, while being limited to production systems. There are also original images that contain hay, and all the mentioned iterations of images were used as input for training the algorithm.
Changes made to improve the manuscript and added sentences: “The background on which the dataset was generated is a white-to-gray gradient, which reduces the complexity of training the AI algorithm and practically represents ideal conditions, however, to achieve the system substandard (such as shadows, non-functional operation of one or more reflectors, or movement of the PCB itself in a non-ideal position), augmentations and processing methods were applied, which is described later in the paper.”
The dataset is not suitable for evaluating the true capacity of YOLOv5.
ANSWER:
This research aimed to examine various models for the detection, localization, and segmentation of the object proposed in the manuscript. Several (four) models were tested, default large and extra large or even bigger models can perform perfectly but is it possible to apply a smaller model for this type of dataset, which is less computationally demanding and has fewer parameters compared to related models? This statement is visible in the manuscript: "The influence of network size and the level of reliability of detection and segmentation of trained models will also be investigated for the problem of objects with similar characteristics at a higher production speed (on the production line in the PCB manufacturing process)."
In addition, there is also a technical application of the given algorithm for the mentioned problems described by the sentence in the manuscript "The goal of this research is to produce an AI model that will satisfy the accuracy requirements with the goal of implementation in robotic systems that use segmentation characteristics to perform the task (in this case, it sorts out, depending on the class of PCBs)." The problem that occurs in production systems with a robotic manipulator, when rotating objects is defined by the sentence in the manuscript: "First is the instance segmentation aspect - most of the research has focused on object detection and localization. This approach does not provide information on the orientation of the device, which can cause issues in instances when the detected objects need to be precisely manipulated, e.g. using an industrial robotic manipulator for sorting [19,20]."
After the research was done, it was shown that for this type of challenge and type of dataset, a smaller model that requires less memory and is faster compared to larger models certainly does the given task perfectly.
In this way, a scientific and technical contribution is obtained that can certainly be applied for real purposes.
I also wanted to comment on the “cross-validation” used in the study. The authors did not adopt a typical training-validation-test data split for model training and evaluation, which is simpler and less biased. Typically cross validation data will be exposed to the model during training, although which is not the case in this study and I need to comment on that too, therefore cross validation results are biased towards superior model performance.
In this study, however, the authors split the entire dataset into five smaller datasets, and trained five different models with the five smaller datasets that were further split into training and test datasets. I’m not sure whether this can be called “cross validation”, as no validation dataset is present and no validation was executed during model training. I believe the authors mistook their test datasets as validation datasets.
ANSWER:
In the paper, a typical five-fold cross-validation was performed, however, during the writing, there was a bad expression, and the authors hereby apologize for the resulting ambiguities. To describe the cross-validation part more precisely, the paragraph is further clarified and updated.
The modifications applied to the manuscript are as follows: " From Figure 3 it is visible that the total data set was first randomly and evenly divided into five parts, i.e. five folds. Then, after dividing the total datasets of the remaining four folds, they are used as a training set, while the initial fold is used as a test set. This ensures that part of the data set does not match the train and the test set and avoids overfitting problems [41].”
TO
“The total dataset is divided into five equal parts (Fold 1 - Fold 5) which makes 20\% of data be used for each fold. Five separate datasets (Split 1 -Split 5) are then created, consisting of one fold, i.e. the test set, and the other four folds that form the training set. For example, for the first split (Split 1), Fold 1, is the test set, while the other four (Fold 2 - Fold 5) are grouped and used for the training set. For the second split (Split 2), Fold 2 is used for the test set while the other folds (Fold 1, Fold 3, Fold 4 and Fold 5) are grouped and form the training set. By applying this procedure to the other three splits, five separate datasets are obtained with the same amount of images (1040 for training and 260 for testing), but with different data (images) for each split separately. By using this methodology, the problem of overfitting is avoided and a more precise evaluation of the AI algorithm used is carried out..”
What is the IoU threshold for identifying TP/TN/FP/FN?
ANSWER
The IoU threshold value for the identification of TP, TN, FP, and FN is 0.6.
The text added in the manuscript is as follows: “Intersection over Union threshold (IoUt) is a parameter that shows the percentage of overlap between the predicted bounding box and the ground truth bounding box. In this research, its value was set to 0.6, i.e. 60 \%. This means that all predicted bounding boxes containing at least 60 \% of the initially labeled image are accepted as true positive predictions.”
Line 133-155 provided unprofessional reasons for not using the entire original dataset. If using the entire dataset is unnecessary and would achieve similar prediction results, which is probably the case, then not using the entire dataset is justified. “Too much work” cannot be a reason for not going above and beyond in research
ANSWER:
The authors hereby apologize for the insufficiently explained description. One of the ideas was to apply the YOLOv5 algorithm to a smaller dataset like this one, in order to clarify the given problem, the following sentence was added in the "Data set description and preparation" section:
The added text is as follows: " It is noticeable that the total dataset is reduced by 6825 images and this is not a coincidence.A CNN-based algorithm has been shown to easily achieve satisfactory performance on large datasets [36]. To investigate whether the model can be trained on a smaller set, and thereby speed up the data collection process in a possible application, a limited dataset is used. For this reason, a condensed dataset is formed that has at least one of the actions shown in Table 1.
.”
Figure 1 should show the images with various dataset generation methods provided in Table 1. The current images in Figure 1 are rather similar to each other and do not seem to be representative
ANSWER:
Based on your comment, Figure 1 has now been changed and the dataset generation methods are more visible. The authors hope that it is now in line with expectations and that it is better presented..
Additionally, Table 1 is unnecessary if the entire original dataset is not used. Instead, only provide information of the shrunk dataset.
ANSWER:
Based on your advice, Table 1 has been modified and the dataset with the difference between the initial and modified data are more precisely described.
The text was additionally modified: "The dataset used in this research consists of 13 classes of PCB microcontrollers and is publicly available on the Kaggle repository [37]. It consists of 1300 images (from the original 8125) that were created by applying different actions such as rotations, translations, centering, etc. Furthermore, the resolution is not consistent and uniform, it consists of different resolutions, and the mean resolution values is 1949 × 2126 pixels by height and width. It is noticeable that the total dataset is reduced by 6825 images and this is not a coincidence. Almost any AI algorithm will have high performance if the dataset has a sufficient amount and quality of data (images) used for training the NN, but in this case a condensed dataset that has at least one of the actions shown in Table 1 was taken into account, the reason for this is a performance test of the YOLOv5 algorithm with a limited dataset of this type.”
Table 2 and Figure 3 and their relevant content are redundant.
ANSWER:
The authors accepted your suggestion and Table 2 and Figure 3 were discarded, although as for the description of Figure 3, the description of cross-validation was left for easier and better presentation of the amount of data, i.e. presentation of the size of the dataset for training AI algorithms. If in the second round of review, the reviewer still requests and considers that a description is not necessary, the authors are ready to adjust the manuscript to improve its quality.
Also to more clearly describe the cross-validation process in subsection 2.1, the manuscript was modified as follows:”The total dataset is divided into five equal parts (Fold 1 - Fold 5) which makes 20% of data be used for each fold. Five separate datasets (Split 1 -Split 5) are then created, consisting of one fold, i.e. the test set, and the other four folds that form the training set. For example, for the first split (Split 1), Fold 1, is the test set, while the other four (Fold 2 - Fold 5) are grouped and used for the training set. For the second split (Split 2), Fold 2 is used for the test set while the other folds (Fold 1, Fold 3, Fold 4 and Fold 5) are grouped and form the training set. By applying this procedure to the other three splits, five separate datasets are obtained with the same amount of images (1040 for training and 260 for testing), but with different data (images) for each split separately. By using this methodology, the problem of overfitting is avoided and a more precise evaluation of the AI algorithm used is carried out.”
Also for Table 2, i.e. the annotation of images, the following modifications of the manuscript were made:”By using the "Smart Polygon Labeling" option, a vastly larger and more precise number of annotation points surrounding the desired object is obtained, as seen in the Figure 2.” in subsection 2.1.“
“Instance segmentation” is the wrong term to be used in this manuscript as there is not multiple instances in each image. “Semantic segmentation” should be used.
ANSWER:
The term “instance segmentation” is changed to “semantic segmentation”.
In addition, in the introduction of this research, a sentence was added that defines it as semantic segmentation instead of instance segmentation:"It is important to emphasize that the term "instance segmentation" is used for algorithms that segment multiple classes on one image, however, in this research, this is not the case because one PCB class is segmented on one image, and it is better to use the term semantic segmentation instead of instance segmentation”
Line 378-380, be very careful with this statement as it is likely incorrect. “Deep networks are less accurate than shallow networks” does not make sense and is likely due to insufficient model training, i.e., too small of a dataset, too simple of a dataset. Deep network performance can plateau to the same level of shallow networks, but it is not because of the network itself, rather, it is because of the dataset.
ANSWER:
The authors agree with your comment, but the results showed that applying similar hyperparameters yields the results presented in the manuscript. Nevertheless, in order to avoid potential confusion and incorrect expression of thoughts, modifications were made in the manuscript: "The conducted research showed that the application of different NN models to this type of dataset gives different results for the same hyperparameters, which indicates that NNs have different properties and thus applications for different types of datasets." and "Also, it is important to note that the given conclusion refers to the challenge presented in section 2.1 and that there is a possibility of deviation of the results from those obtained in a different environment than in the current dataset." in the conclusion section.
We hope you will be satisfied with our answers to the questions posed. We have marked the changes made to the manuscript due to your comments using blue highlight.
Kindest regards,
The Authors
Reviewer 3 Report
- The motivation in the abstract needs further elaboration. What is the problem with making and sorting PCBs on lines 2-3?
- The goal of the research should be moved before the literature review in the introudction.
- The objectives of the paper on lines 84-86 need rephrasing and elaboration. What is the input? and what is the output?
- Lines 22-27 capitalize the first letter of the first word of the sentence. The same in lines 95-98.
- Dataset not data set .
- The statement "and the images were selected randomly to eliminate the possibility of a poor, low-quality data set selection" on line 136 is self contradictory.
- Did the augmentation in Table 5 increase the size of the dataset? If so then the results will be misleading as data leaking will be rampant.
- What was the value of the IoU used to generate the results?
- What were the backbone CNNs and detections heads locations?
- Similar studies utilizing Yolo can be cited so that to established the trustworthiness of the models, see Detection of K-complexes in EEG signals using deep transfer learning and YOLOv3. Cluster Comput (2022). https://doi.org/10.1007/s10586-022-03802-0
- The table of abbreviations as required by the journal template is missing.
Author Response
Respected Reviewer,
thank you very much for your review of our manuscript. We have tried our best to respond to the issues you have noted in your manuscript. Please find our responses below. Changes in the manuscript made due to your comments have been marked red.
The motivation in the abstract needs further elaboration. What is the problem with making and sorting PCBs on lines 2-3?
Robotic manipulators have a high rate of use in factories more precisely in object sorting. The problem arises with the orientation of the object, i.e. how and in what way the robotic manipulator will grasp the object and perform the "pick and place" action.
To better explain the motivation of the work, a sentence was added in the abstract of the manuscript: "One of the main challenges in factories that use robotic manipulators for "pick and place" tasks is object orientation, because the robotic manipulator can misread the orientation of the object and thereby grasp it incorrectly, and for this reason, object segmentation is the ideal solution for the given problem."
The goal of the research should be moved before the literature review in the introudction.
The goal of the research was moved before the literature review and slightly modified, i.e. instead of the original "The goal of this research..." it was reformulated into "With that in mind, the goal of this research..."
The objectives of the paper on lines 84-86 need rephrasing and elaboration. What is the input? and what is the output?
ANSWER:
Lines 84-86 are moved based on your previous comment before the literature review. Furthermore, the objective of this paper is to obtain the class, location, and segmentation of PCBs based on images captured by camera or some other image-recording devices.
To further clarify the part that was moved before the literature review (the main objective with input and outputs of the AI algorithm), several modifications to the text are applied: "With this in mind, the goal of this research is to produce an AI model that, based on input data (images) obtained from a camera, can acquire output information about the location, class, and segment of the object, in this case, PCBs. At the same time, it must meet the high-performance requirements with the aim of implementation in the robotic systems that use segmentation characteristics to perform the task (in this matter it is sorted depending on the PCB class)."
Lines 22-27 capitalize the first letter of the first word of the sentence. The same in lines 95-98
ANSWER:
Lines 22-27 and 95-98 have now been rewritten and the beginning of the sentence begins with a capital letter.
Dataset not data set
ANSWER:
The term "data set" has been changed to "dataset".
The statement "and the images were selected randomly to eliminate the possibility of a poor, low-quality data set selection" on line 136 is self contradictory.
The authors agree with the statement, and the sentence was modified, i.e. removed, for the reason that there were modifications at the request of the previous reviewer, and your comment was taken into account when writing.
The changes are in lines 122-138 and are not additionally indicated for less complexity of reading during the review.
Did the augmentation in Table 5 increase the size of the dataset? If so then the results will be misleading as data leaking will be rampant
The description of cross-validation is described in subsection 2.1, initially the total (modified i.e. reduced) dataset was divided into five splits. The test fold is separated from the training folds, and after the division the training set was automatically augmented and described in the manuscript. To answer your question. The dataset was augmented, but only in a training set after dividing the data into five splits, as described in the manuscript.
To emphasize the augmentation process, the text above Table 4 was added: "Also, it is important to emphasize that the YOLOv5 algorithm implements the augmentation of images in the training process. This means that based on the data (images) found in the training set, augmentation is performed on them, while the test set remains untouched, provided that the data division was initially performed, which was done and described in the subsection 2.1. This approach avoids the problem of "data leaking", i.e. repetitions of almost identical data in the train and test set.”
What was the value of the IoU used to generate the results?
The Intersection over Union for generating results was set to 0.6. At the request of the previous reviewer, it was added to Table 2 and further described. It is not indicated in red but in blue to make it easier to read when reviewing.
What were the backbone CNNs and detections heads locations?
The backbone is located immediately after the input image, which is used for feature extraction, after feature extraction, feature aggregation is carried out using the neck, and finally, the predictions are reduced in the head. Furthermore, the authors would like to thank you for the suggested quote, because the description of the YOLO structure is additionally presented.
Additionally, the following text was added to the manuscript: "The backbone is located right after the input image and it serves for features extraction, after that the data is processed with the neck of the CNN which performs resolution feature aggregation and finally the final predictions based on the resolution of the object are generated in the head [47,48].”.
The table of abbreviations as required by the journal template is missing.
The table of abbreviations is added at the end of the manuscript.
We have used red highlight to note the changes made to the manuscript due to your comments. We hope you will be satisfied with our answers to the questions posed.
Reviewer 4 Report
The paper is describing approach based on new YOLOv5 and on base of experiments authors have shown that the solution requires quite a modest amount of resources along with achieving really high target labelling precision.
The paper looks generally well organized (background, methods, methodology, experiments and results, discussion) and written in a good language (only one typing error in Table 3 - parameter "Patiance".
In Table 2 (Example of labeling images) it remains unclear what exactly dot values represent (with 13 decimal places after comma precision).
Appendix A is unproportionally large. To explain labelling, some other form of presentation should be used.
Unfortunately, nor the title either abstract either introduction are not opening clearly what problem is solved. Is the problem of defecting erroneous PCB-s or sorting them after production or during disposal (when initial natural order of PCB-s is lost), or are the objects rather full modules (PCB along with all components)?
The task of PCB is not only provision of interconnect (line 20) but to guarantee signal quality, timing, also to form electronic components like inductors, antennas (e.g. WiFi) etc.
I would also argue that using ML methods regarding PCB-s has been a research problem over multiple decades, not only recent years. Attempts to combine NN-s, fuzzy logic etc were done in 80-thies already. Deployment of DNN in this domain might be indeed topic of recent years.
Figure 6 does not present performance enough well, it is kind of out-of-scale situation. One solution might be separation of std and mean bars and present them using adjusted scale which is emphasizing small differences.
Results obtained using different YOLO models are so close that more important are required memory space (in Table 10) and processing speed.
This is another drawback - experiments do not carry timing information which would be valuable aspect for reader of the paper.
Author Response
Respected Reviewer,
thank you for your detailed review of our submitted manuscript. The response and answers to the questions you posed are below. Changes in the manuscript made due to your comments have been marked cyan.
In Table 2 (Example of labeling images) it remains unclear what exactly dot values represent (with 13 decimal places after comma precision)
In Table 2 there was an example of polygonal labeling. The point of that example was to show the practicality of using the "Smart Polygon" options, in translation for example when the user manually marks usually marks about 20 dots on average, which takes a lot of time on bigger datasets. Using this option results in 235 labeled dots for segmentation where you only need to label the entire object with one rectangle and Roboflow's AI will automatically label the whole with a large number of dots (in this case 235) showing exceptional precision.
However, in this case, Table 2 was removed at the request of the previous reviewer, and thus the ambiguity of the presentation of Table 2 in the manuscript was removed.
Appendix A is unproportionally large. To explain labelling, some other form of presentation should be used.
Appendix A contains examples of real PCB classes and predicted PCBs using the resulting YOLOv5 model. Given that there are 13 classes of PCBs, the authors decided to move it from the Results to the Appendix section. This example was not about labeling the images but the ground truth of the PCB class and predicted class. To clarify the issue, the text in the Appendix section has been changed to:”In this part of the Appendix section, a presentation of the real class of PCBs (ground truth) is given, and a predicted class using the YOLOv5 model is shown.”
In case the reviewer does not agree with this way of presenting the results, the authors are willing to present the solutions obtained in this research in a completely different way in the second round of reviews.
Unfortunately, nor the title either abstract either introduction are not opening clearly what problem is solved. Is the problem of defecting erroneous PCB-s or sorting them after production or during disposal (when initial natural order of PCB-s is lost), or are the objects rather full modules (PCB along with all components)?
ANSWER:
The goal of this research was to implement the YOLOv5 algorithm model, which with its precision and speed will qualitatively sort one of the 13 classes of PCBs. Furthermore, the emphasis is that the order of the PCB class is unimportant, i.e. that the algorithm perfectly recognizes the PCB class in a short time. To emphasize the application of this algorithm, the following text was added in the introduction of this manuscript:” Obtaining a high-quality model trained with the YOLOv5 algorithm creates an opportunity to solve the problem of uneven sorting i.e. multiple classes independent of the order on the production line. The robotic manipulator will almost perfectly recognize any of the 13 classes of PCBs and, based on the recognized class, place them in the designated section, regardless of the PCB class order on the conveyor belt. This speeds up and improves the sorting process and the sorting problem with a very simple implementation m into the robotic system.
The task of PCB is not only provision of interconnect (line 20) but to guarantee signal quality, timing, also to form electronic components like inductors, antennas (e.g. WiFi) etc.
ANSWER:
The authors agree with your statement and some more functions have been added and added to the manuscript in the introduction section.
The added text in the introduction section is as follows:" guarantees signal quality, timing, and many other valuable functions, and for this reason, it can be found as a part of:”
In case it is necessary to add more examples of what a PCB consists of, the authors will subsequently add more examples of what a PCB consists of in the next review, although we believe that it is not necessary to describe the composition of PCBs in detail, the core of the research is the application of artificial intelligence algorithms for detection, classification, and segmentation of PCBs for implementation in robotic manipulators.
I would also argue that using ML methods regarding PCB-s has been a research problem over multiple decades, not only recent years. Attempts to combine NN-s, fuzzy logic etc were done in 80-thies already. Deployment of DNN in this domain might be indeed topic of recent years.
ANSWER:
The authors agree with the given comment, and the sentence in the introduction of this paper has been reformulated from: "The use of image-based machine learning (ML) methods for various tasks regarding PCBs is a topic of numerous research in recent years." in "The use of image-based machine learning (ML) methods for various tasks regarding PCB has been the subject of numerous research in the past several decades."
Figure 6 does not present performance enough well, it is kind of out-of-scale situation. One solution might be separation of std and mean bars and present them using adjusted scale which is emphasizing small differences.
Figure 6 is divided into two parts, one part shows the mean values of individual metrics (in the new manuscript Figure 5), while the second part shows the standard deviation (Figure 6). The displayed actions with minor modifications to the description are shown in the discussion section.
Results obtained using different YOLO models are so close that more important are required memory space (in Table 10) and processing speed.
This is another drawback - experiments do not carry timing information which would be valuable aspect for reader of the paper.
ANSWER:
The authors agree with the given comment, and Table 9 has been additionally modified, i.e. the parameter "Mean inference speed in milliseconds [ms]" has been added and is additionally described in the manuscript: "Besides the memory and computational requirements, one of the most important feature of the YOLOv5 algorithm is the inference speed in which YOLOv5 excels for each NN model. The inference time is calculated on the validation set (data that was not in the test set) and the mean value was taken for all five splits under the conditions that the IoU threshold was set to 0.6, while the confidence threshold was set to 0.9. As shown in Table 9 YOLOv5 has the lowest inference time using the YOLOv5s model. The YOLOv5n model is slower by 1 ms than the YOLOv5s model, while YOLOv5m is slower by 5 ms and YOLOv5l by as much as 9 ms. Regardless of the time difference, each model is within acceptable limits in terms of inference speed, which further confirms the possibility of applying the YOLOv5 algorithm AI for a given challenge."
We have used cyan highlight to note the changes made to the manuscript due to your comments. We hope you will be satisfied with our answers to the questions posed.
Kindest regards,
The Authors
Round 2
Reviewer 2 Report
The authors have addressed my comments. The manuscript in its current condition is publishable. I would encourage the authors to double check the language and grammar of the manuscript, since I noticed an error in line 417-418 “the influence … was carried out”. Also the conclusion section reads like discussion to me. I would advise keeping the section concise and to the point, as the section is about “conclusions”.
Author Response
The authors want to thank the reviewer for his comments and suggestions. The language and grammar were double-checked, the aforementioned error on lines 417-418 was corrected, and the conclusions section was made as concisely as possible.
Reviewer 3 Report
The authors answered my comment.
Author Response
The authors want to thank the reviewer for his time and effort and for constructive comments and suggestions which greatly improved the manuscript's quality.
Reviewer 4 Report
Thanks for you work! Given remarks and suggestions about the first version has been well taken into account and questioned decisions (e.g. appendix related) reasoned. The paper has been improved (its readyness for readers) remarkably.
Author Response
The authors want to thank the reviewer for his time and effort with his comments and suggestions which greatly improved the manuscript's quality.